# Single Quasi–Symmetrical LED with High Intensity and Wide Beam Width Using Diamond–Shaped Mirror Refraction Method for Surgical Fluorescence Microscope Applications

**DOI:** 10.3390/diagnostics13172763

**Published:** 2023-08-25

**Authors:** Minki Ju, Kicheol Yoon, Sangyun Lee, Kwang Gi Kim

**Affiliations:** 1Medical Devices R&D Center, Gachon University Gil Medical Center, 21, 774 beon-gil, Namdong-daero Namdong-gu, Incheon 21565, Republic of Korea; sprjk0504@gmail.com (M.J.); kcyoon98@gachon.ac.kr (K.Y.); l0421h@gmail.com (S.L.); 2Department of Biomedical Engineering, College of Health Science & Medicine, Gachon University, 1342 Seongnamdaero, Sujeong-gu, Seongnam-si 13120, Gyeonggi-do, Republic of Korea; 3Department of Health Sciences and Technology, Gachon Advanced Institute for Health Sciences and Technology (GAIHST), Gachon University, 38-13, 3 Dokjom-ro, Namdong-gu, Incheon 21565, Republic of Korea

**Keywords:** LED, fluorescence emission, quasi–symmetrical LED, emission power, beam mirror

## Abstract

To remove tumors with the same blood vessel color, observation is performed using a surgical microscope through fluorescent staining. Therefore, surgical microscopes use light emitting diode (LED) emission and excitation wavelengths to induce fluorescence emission wavelengths. LEDs used in hand–held type microscopes have a beam irradiation range of 10° and a weak power of less than 0.5 mW. Therefore, fluorescence emission is difficult. This study proposes to increase the beam width and power of LED by utilizing the quasi–symmetrical beam irradiation method. Commercial LED irradiates a beam 1/r^2^ distance away from the target (working distance). To obtain the fluorescence emission probability, set up four mirrors. The distance between the mirrors and the LED is 5.9 cm, and the distance between the mirrors and the target is 2.95 cm. The commercial LED reached power on target of 8.0 pW within the wavelength band of 405 nm. The power reaching the target is 0.60 mW in the wavelength band of 405 nm for the LED with the beam mirror attachment method using the quasi–symmetrical beam irradiation method. This result is expected to be sufficient for fluorescence emission. The light power of the mirror was increased by approximately four times.

## 1. Introduction

Because malignant tumors with high invasiveness have high metastasis, the goal of surgery is to reduce the risk of recurrence by completely resecting the tumor during cancer surgery. Malignant tumors are distributed with numerous blood vessels [1].

The color of the tumor and blood vessels are nearly identical [2], making it difficult to distinguish between them. In addition, it is challenging to resect tumors that have the same color as blood vessels. The risk of recurrence will increase if blood circulation disorders are caused by damaging blood vessels during the tumor resection process or if fewer tumors are removed to prevent blood vessel damage. Therefore, fluorescent staining is performed to observe the state of tumor removal and blood circulation of blood vessels [3,4,5,6]. In addition, the state of fluorescently stained tumors and blood vessels can be observed using an operating microscope through the color of the fluorescent wavelength band [7,8].

In addition, fluorescence–emission–guided tissue observation can detect small and numerous lymph nodes. This approach offers the advantage of swiftly observing these nodes during the surgical procedure [9,10].

Important elements of surgical microscopes include optical and light–emitting systems that ensure viewing angles (*θ*) for observation and light intensity suitable for creating long working distances (WD) and providing sufficient surgical space for surgeons while monitoring fluorescent images [11]. Surgical microscopes have been studied in order to meet these technical requirements [12,13,14].

For example, the fluorescence–emission–guided lesion observation method using near–infrared (NIR) rays uses a NIR window (650–1350 nm) to separate the light source irradiation wavelength band and the fluorescence emission wavelength band and induces the camera to pass and capture only the fluorescence emission wavelength band [15,16]. This method can improve the quality of monitoring images for observation. A case of developing an integrated imaging system by combining NIR fluorescence and gamma–positron imaging has been reported, which can effectively observe lesions using the fluorescence–emission–guided method and simultaneously observe lesions through radiography [17,18]. Single or combined imaging monitoring methods, such as Cerenkov luminescence imaging, optical coherence tomography, photoacoustic imaging, Raman spectroscopy, thermography imaging, chemiluminescence imaging, confocal microscopy, and fluorescence lifetime imaging, have also been developed. This method enables real–time image observation of the nuclear–NIR imaging system during surgical procedures and can be used for surgical lesion observation [19,20]. A case of research on a system design capable of multiple fluorescence emission using LEDs with NIR and infrared (IR) wavelength bands is reported. The advantage of this is that various observations can be made on the state of blood flow in blood vessels, the state of tumor removal, and the detection of lymph nodes using indocyanine green, 5–Aminolevulinic Acid (5–ALA), and fluorescence sodium [21]. However, surgical microscopes that use lasers, xenons, and halogen bulbs as light sources provide sufficient fluorescent light even when the WD is large. However, the light sources of these surgical microscopes can negatively affect human safety [22]. Additionally, the laser is straightforward and has a narrow beam width, which reduces the viewing angle and range of fluorescence expression guidance for the lesion [23]. Moreover, it requires large amounts of energy for operation, leading to increased carbon emissions [24]. The substantial energy consumption also releases a significant amount of heat, which can cause the light source module used in surgical microscopes to break down quickly. Therefore, there are several restrictions on obtaining approval for medical devices. To compensate for these disadvantages, the number of observation tools using light–emitting diodes (LED) as light sources is increasing [25,26].

LEDs do not negatively affect the human body. Moreover, they consume less energy, which makes them mechanically durable. Additionally, because the irradiation area is relatively wide, they provide a wider range of lesion fluorescence expression [27]. However, the emission intensity of LEDs is weaker than that of a laser; therefore, the fluorescence emission is weaker. Moreover, the range of light irradiation is wide, making it difficult to adjust the observation areas. Furthermore, because the intensity of light is uneven, bright and dark parts occur together, which may interfere with the observation field during surgery, hindering an accurate diagnosis. To overcome these problems, this study proposes a single quasi–symmetrical LED beam irradiation method that can expand the beam width of LEDs and increase their power. The light was reflected using four mirrors to create a quasi–symmetrical LED beam, which was then collected at the target point. Compared with the existing methods, the irradiation intensity of the LED can be increased by 4.0 times, and the beam width can be increased by 4.3 using the proposed method, resulting in a more intense and evenly spread light beam.

## 2. Method for Increasing the Beam Width and Emission Power

### 2.1. Examining the Limit of Fluorescence

Following the intravenous injection of a fluorescent contrast agent into the tumor, as illustrated in Figure 1a, the fluorescent contrast agent stained on the tumor becomes fluorescent (wavelength λ_ext_, power P_ext_) due to chemical changes. A surgical microscope (or camera) captures a color suitable for the fluorescence expression wavelength (λ_em_), such that the tumor appears green or yellow on the external monitor.

A camera with excellent performance is recommended for the clear observation of fluorescent tumors on an external monitor. Additionally, for the contrast medium to exhibit sufficient fluorescence, it should be irradiated with high–intensity light. 

Conventional surgical microscopy requires a minimum power of 200 mW [27]. Furthermore, as shown in Figure 1b, considering working distance (WD) and air transmission loss, sufficient fluorescence is generated when the target is exposed to a light beam with a minimum power of 0.5 mW [28]. Currently, small and light handheld surgical microscopes are required [27]; however, commercial LED products suitable for handheld surgical microscopes can only have a maximum power of 18 mW due to their physical limitations and fabrication characteristics. If an LED with an output power (P_LED_) of 18 mW is used with a single symmetric method (irradiation angle, *θ*: 0°), as shown in Figure 1b, loss (58% @ WD of 20 cm) occurs, as described in Equation (1), which decreases the power (*P_o_*) of the LED, as shown in Figure 1c [29,30]. Therefore, the maximum power analyzed by the target is approximately 8.0 pW, as described in Equation (3), and fluorescence expression becomes impossible because of low emission power.
(1)$$P_{o}=\frac{2P_{max}}{{2\pi \varepsilon_{r}r}^{2}}e^{r^{2}}$$
(2)$$P_{l}=\frac{I_{o}P_{o}}{2{\pi \varepsilon_{r}r}^{2}}$$
(3)$$I_{o}=\frac{P_{max}}{4{\pi \varepsilon_{r}r}^{2}}$$
where *I_o_* denotes the intensity of light emitted from the LED into the air, *P_max_* is the maximum power generated inside the LED module (the power of light generated during the electron movement process in the semiconductor), *W* denotes the bandwidth of the emitted light of the LED, *r* denotes the slope at which the LEDs light is emitted up to the maximum distance, *ε_r_* denotes the dielectric constant in the air (*ε_o_* = 8.858 × 10^−12^) [31], and *P_o_* is the power emitted from the LED. The irradiated light intensity did not reach the minimum reference point for fluorescence expression, making fluorescence expression impossible. The range of the beam irradiation angle (*θ*) of the LED is 10°; therefore, the tumor observation viewing angle appears to be wide. However, the LED beam is not irradiating the entire lesion, as shown in Figure 2; therefore, the fluorescence–guided observation is limited.

If the intensity of light (*I_o_*) of optimized power is irradiated using a single asymmetrical method (irradiation angle: 15°), as shown in Figure 1b, the range of light irradiation and the viewing angle is widened [32]. However, because the intensity (*I_o_*) of light, distance (*r*), and power (*P_o_*) are not uniform, the brightness of the fluorescence expression may also be different because of the strong (−x_1_ to x_0_) and weak (x_0_ to x_1_) parts. Therefore, because the bright and dark parts of the lesion are divided during fluorescence diagnosis, observation and diagnosis are hindered, and fluorescence expression may be difficult because the light irradiation becomes uneven. In addition, because the asymmetrical method requires a relatively long distance between the light source and the target, power (*P_o_*) is decreased owing to loss (*l*) in the air, and fluorescence expression in tumors becomes difficult. Therefore, the diagnostic view has limitations.

As shown in Figure 1b, if dual (LED_1,2_) or multi–asymmetrical (LED_1,2,3_) LED sources are used, the intensity of the light (*I_o_*) increases [27]. However, commercialization may be difficult owing to the complicated configuration of the device and increased unit cost and power consumption [33]. Moreover, the light source may become physically destroyed because of the heat generated by the increased power consumption.

### 2.2. Quasi–Symmetrical Single LED Divergence

For conventional LEDs, it was assumed that light (intensity *I_o_*) is irradiated at the *op* point, as shown in Figure 3a. In this case, when light (*I_o_* = i_1_ to i_n_, −i_1_ to −i_n_) reached d_j0_ through m (m_u_, m_c_, and m_l_), the intensity (*I_o_*) of the light was uniformly distributed from d_j0_ to d_j−n_ (or d_jn_), as described in Equations (4) and (5).
(4)$$x=2ytan\left(\theta /2\right)$$
(5)$$\theta =2{tan}^{-1}\left(x/2y\right)$$

Additionally, if WD increased, the light beam was wider. The viewing half angle (*θ*) between d_j−n_ and d_jn_ was 10° (Thorlabs LED 405E, with output power (*P_o_*) of 18 mW at a wavelength of 405 nm). However, the power (*P_o_*) of the light differs from its irradiation intensity (*I_o_*). 

The intensity of the light emitted from the op was different from that of d_j0_ and d_j1_. The power (*P_o_*) of d_j0_ decreases when it reaches d_jn_ (or d_j−n_). Therefore, the intensity (*I_o_*) of light is different, resulting in differences in the fluorescence expression. Specifically, as WD increases, the intensity of the light source is absorbed in the air, resulting in power loss. Therefore, power *P_o_* is reduced by 1/e^2^ and, eventually, the fluorescence expression is weakened owing to the loss of power (*P_l_*). Moreover, because the beam width (10°) of the LED is narrow, fluorescence is partially expressed only in the area where light reaches; therefore, the fluorescence emission–guided observation field is narrowed, which hinders diagnosis. Conventional LEDs use a symmetric method, yet an asymmetric method is more effective for fluorescence expression. However, as light is reflected diagonally, it is divided into bright and dark parts, and the tone of the irradiated part varies.

In particular, if the oblique line of the beam becomes longer (an increase in WD), the beam intensity and fluorescence expression decrease, hindering lesion observation. For the beam divergence of an LED, the proposed quasi–symmetric method is more suitable. In the quasi–symmetric method, the LED beam pattern appears symmetric, as shown in Figure 3b. When a typical LED diffuses from op to d_jn_ (or d_j−n_), as shown in Equations (4) and (5), the intensity (*I_o_*) of light is transmitted from mu to *m_l_* without reflection or refraction [33]. In contrast, the quasi–symmetric method proposed in this study uses m_u_, m_c_, and m_l_ to reach d_j−n_ (or d_jn_) when the light (intensity: *I_o_*) of the LED starts at op. In this case, *I_o_* emitted from op linearly reached up to m_u_ (*I_u_*), and I_o_ was refracted by (*θ*) 45° through m_c_, as described by Equations (6) and (7).
(6)$$P_{o},\;P_{l},P_{u}=\theta I_{o}$$
(7)$$I_{l},\;I_{u}=\sqrt{2}WD$$

The light was refracted 45° and traveled (*I_l_*) from *m_l_* to d_j_ (d_j0_ to d_jn_). Therefore, the beam pattern appeared to maintain a three–dimensional diamond shape (octahedron). When the beam is distributed from *op* to d_jn_, conventional LEDs have a wide light irradiation range, but the intensity (*P_o_*) is the strongest and brightest at d_j0_, and the intensity (*I_o_*) gradually decreases, as described by Equations (8)–(10), as the light moves from d_j1_ to d_jn_ (or d_j−n_) [34]. However, in the quasi–symmetric method, the light generated from *op* is concentrated on m_c_. The light *I_l_* concentrated in mc may spread evenly from d_j0_ to d_jn_ (or d_j−n_) through refraction.
(8)$$\vec{I_{o}}=\vec{I_{u}}-2\vec{I_{l}}\;\left(\vec{I_{o}}\cdot \vec{I_{u}}\right)$$
(9)$$\vec{I_{o}}=\vec{I_{l}}-2\vec{I_{u}}\;\left(\vec{\theta_{u}}\cdot \vec{\theta_{c}}\right)$$
(10)$$I_{l}=I_{o}exp\;\left(\frac{2{WD}^{2}}{{P_{o}}^{2}}\right)$$

It is analyzed that the intensity (*I_o_*) of the light directly reaching d_j0_ is constant, the intensity of the light refracted from m_c_ is distributed uniformly by the beam width from d_j0_ to d_jn_ (or d_j−n_), and the intensity (*I_o_*) of the light directly irradiated from *op* is N times higher than that of a single LED. Therefore, using the quasi–symmetric method, sufficient light intensity can be created for fluorescence expression while maintaining a wide beam width.

## 3. Experimental Results

As illustrated in Figure 4, this experiment compared conventional LED irradiation with a quasi–symmetrical LED irradiation method using mirrors. Figure 4a shows the device (3D modeling layout and fabrication structure) for measuring the brightness and power of light using conventional LED illumination. The LED was fixed to a clamp (Manfrotto 244N, Cassola, Italy) to measure the conventional LED (Thorlabs M405L2–C1, Newton, NJ, USA). Figure 4b shows the device (3D modeling layout and fabrication structure) used to measure the brightness and power of light through quasi–symmetrical LED illumination. The 3D modeling layout was designed using the SolidWorks tool, and the structure and form were designed using 3D printing technology.

The proposed quasi–symmetrical LED beam method places up to four mirrors (Uniart 2500 safety coated mirror #4) of rectangular shape around one LED (Thorlabs M405L2–C1). A power meter (Thorlabs S121C) was installed at the bottom to compare the power (*P_o_*) of light measured using the two methods. Additionally, lux meters (Thorlabs S142C, Sincon ST–126) were used to conduct comparative experiments on the brightness when measuring the LED power, and a spectrometer (Ocean Optics Flame spectrometer, FLAME–S) was used to verify the wavelength bands of the LEDs. Figure 4c shows the setup to measure the brightness and power of light under conventional LED irradiation and quasi–symmetrical LED irradiation. Figure 4d shows the experimental setup for measurement using a different number of mirrors (#1 to #4).

As shown in Figure 4a, the WD of the conventional LED is fixed 20 cm above the target. As shown in Figure 5, the beam irradiation angle (θ_b_) is 43°, the wavelength is 405 nm, and the beam power reaching the target is 0.150 mW. Moreover, the brightness of the light is 10,785 Lux. Therefore, a power of 0.150 mW may satisfy the condition of fluorescence expression.

Conventional LEDs have the characteristics of symmetric beam irradiation, and as WD increases, the beam width increases, whereas the intensity of light reaching the target decreases. As shown in Figure 4b, the quasi–symmetric LED beam illumination method has an LED fixed to the center of the upper end, and four rectangular mirrors are installed around the LED. The distance (W_Dom_) from the LED (*op*) to the mirror center (m_irw_) is 2.95 cm. The size (a × b) of the mirror is 5.0 × 5.5 cm^2^, and the distance (c) between the mirrors facing each other is 5.9 cm.

If the light of the LED (*op*) is irradiated with a WD of 20 cm from above the target point, the light reaches mirrors (m_u_), as shown in Figure 3 and Figure 4b. Subsequently, the light reaches the target through 45° refraction from the m_c_ of the mirror, as shown in Figure 3b. The power (*P_o_*) reaching the target of the quasi–symmetric LED is 0.60 mW, with a wavelength of 405 nm, as shown in Figure 6. The irradiation angle (θ_m_) is 43° and the brightness of light is 38,821 lux. Therefore, a power (*P_o_*) of 0.60 mW may satisfy the condition of fluorescence expression.

When measuring the power using the power meter, the results were obtained through experiments based on the fact that the power changes according to the number of mirrors. Table 1 and Figure 7 show the results of measurement with varying the number of mirrors. In this case, as shown in Figure 7, the power increases proportionally as the number of mirrors increases. Therefore, four square mirrors with minimal refractive losses were used to obtain the maximum power.

As shown in Figure 8, an experiment was conducted to test the possibility of fluorescence expression using the quasi–symmetrical LED irradiation method and compare it with the conventional method. For the experiment, samples of fluorescent contrast agents (Fluorescite inj. 10% [500 mg] @ Alcon, Seoul, Republic of Korea) were used. Figure 8a,b show the results of the experiment on the fluorescence expression using the conventional (symmetrical LED) and the proposed (quasi–symmetrical LED) methods, respectively. To compare the results of fluorescence expression, a fluorescence emission photograph using NIR cameras (Lt–225c, lumenera camera, and TELEDYNE, Thousand Oaks, CA, USA) is presented.

## 4. Discussion

The method proposed in this study differs from the conventional LED irradiation method because it can be viewed as an asymmetrical or symmetrical method. The refraction and inverse refraction of the beam occurs simultaneously through the mirrors, and the intensity of light changes, creating the appearance of an asymmetrical irradiation method. However, it can also be observed as a symmetric irradiation method with regard to the uniformity of light intensity. Therefore, it is defined as a quasi–symmetrical irradiation method. The value of WD was fixed at 20 cm because it should be a minimum of 20 cm for a surgical microscope to provide surgeons with enough space for operating a fluorescence–guided surgery [35].

In this experiment, a total reflection mirror must be used to facilitate fluorescence expression. Generally, mirrors can be of two categories: normal mirrors and total reflection mirrors [36]. As a normal mirror induces refraction (>0.35%) because of its thickness, an error resulting from unintended refraction occurs. In contrast, the total reflection mirror has negligible refraction (<0.35%) inside the mirror; therefore, the LED light can be reflected at the desired angle [37,38]. LED light can also be collected, and the beam width can be maintained using concave or convex lenses [39]. However, such a method has a disadvantage in that the light of the LED cannot be adjusted to an intended angle. When a mirror is used, the height of the mirror can be controlled to adjust the beam width and the direction of light.

Furthermore, the experimental equipment was blackened to increase the efficiency of the light source by absorbing unnecessary light from the outside, ensuring that the irradiated light source has no interference from the outside. If the LED is extremely close to the target (<10 cm), the light intensity increases (>2 times); however, the beam width becomes narrow (inversely proportional to WD), and light reflection may occur due to strong light [33]. When lesions are photographed using LEDs in the process of diagnosing fluorescence staining in the operating room, the intensity of LED light may increase, resulting in specular reflection. As specular reflection is photographed by a camera, it may block the lesion view and interfere with diagnosis [4,5,6,12,13,14]. However, if the WD of the LED becomes more than 20 cm, more energy (proportional to distance) will be consumed [24] for fluorescence during the fluorescence diagnosis process, thus generating more heat and leading to physical destruction [27]. If the LED is operated with extra–low energy (<18 mW), fluorescence expression becomes difficult, leading to difficulties in diagnosis. Considering this trade–off situation, the value of WD was fixed at 20 cm.

When a beam expander or splitter is used, the laser intensity is stronger than that of the LED, and the beam width (θ_w_) will be increased; however, light energy is lost owing to parasitic resistance inside the expander or splitter, thereby reducing efficiency and generating a slight shadow area [26]. Considering this phenomenon, this study could maintain beam width and maintain power four times higher than conventional LEDs by using one LED and four full–reflective mirrors. 

Table 2 lists the comparison results for the proposed and existing methods [32,40,41,42,43]. Table 3 compares the proposed method with the previous method while matching their WD values. As a result of analyzing the power reaching the target in the table, while applying the dielectric constant of air (ε_r_) by substituting Equations (1)–(3), when light generated from the LED reaches the target point through the air layer, most of its power is absorbed by the air, which results in the power (*P_o_*) reaching the target to be decreased by more than N times. The 1_2_ asymmetrical LEDs (TLHB5400) are configured in a ring–type array to increase power, and the application of Halogen lighting (NS6B083T) is significantly low in power despite their high input power [32,43]. Although the laser has a higher power than the proposed method, the power loss reaching the target is extremely large [40,41,42,43] owing to the power that is absorbed by the air. In the case of LEDs, the existing method has a shorter WD than the proposed method. Therefore, assuming the same WD as the method proposed in this study, as listed in Table 3, the existing method has a lower power than the proposed method, and cannot satisfy the minimum conditions for fluorescence emission [32,43].

Laser meets fluorescence emission conditions owing to its strong intensity, yet it is subject to numerous restrictions in medical system applications because it is harmful to the human body [25]. Moreover, as the laser has a beam width of 42° or less, there is a limit to irradiating the entire lesion, as shown in Figure 2. In contrast, most LEDs suitable for the licensing process in medical devices have a wide beam width, but they cannot irradiate the entire lesion with the intensity required for fluorescence emission. It is analyzed that the proposed method used in this experiment generates sufficient power to satisfy the conditions of fluorescence expression and has a wide beam width (>40°), thereby enabling the entire lesion to be irradiated.

Some additional items to be considered are the need to analyze the characteristics of beam irradiation and change according to the number of mirrors, the shape of the mirrors (circular or square), and the arrangement of the mirrors (rectangular, triangular, or cylindrical). As for the structure of the mirror, when the LED beam is irradiated, all four mirrors must be filled with four quadrants, as shown in Figure 9. Therefore, the beam must be irradiated in all four quadrants (four mirrors) so that the intensities of the reflected beam in the four quadrants are widely irradiated to the lesion. If there are three mirrors because there is no one mirror, the intensity of the light source will be lost because the light beam is not reflected in the part where there is no mirror, as shown in Figure 9. Even if there are more than four mirrors, all four quadrants must be filled with mirrors; therefore, using more than four mirrors only increases the unit price. As shown in Figure 9, if a circular mirror is used, the horizontal (h) and vertical (v) sizes of the mirror cannot be freely adjusted. The increase in the irradiation intensity of the beam should also consider the individual areas for the horizontal (h) and vertical (v) sizes of the mirror. As shown in Figure 9, the circular mirror cannot individually adjust the horizontal (h) and vertical (v) areas. In addition, if circular mirrors are connected, as shown in Figure 9, a space (sp) between the circular mirrors must exist. This void (sp) will cause an intensity loss of the LED beam because the LED beam cannot reflect. However, square mirrors can be connected continuously without gaps, as shown in Figure 9, to fill the quadrants with mirrors.

In addition, if configured in a triangular shape rather than a quadrangle, the beam irradiation area is narrowed; therefore, the beam irradiation width is also narrowed. In the case of a cylindrical shape, it is difficult to individually tune the horizontal and vertical dimensions of the mirror; therefore, beam intensity control is limited. In addition, the amount of intensity of beam irradiation is changed. This is because the size of a circular mirror and a cylindrical mirror have the property of a circumferential ratio in which the horizontal and vertical sizes (R) of the mirror size or the area of the beam irradiation space increase together. In addition, the amount of intensity of beam irradiation is changed. The main purpose of this proposed method is to secure the observation field of the lesion by widening the width and range of beam irradiation in a state where all four quadrants are filled with mirrors.

Most of the systems used in medical fields have large equipment and high–beam intensity of light sources [27]. However, large equipment is expensive, difficult to maintain, and has a limited range of beam irradiation directions. Therefore, the monitoring field is limited because the lesion observation range is limited [27]. In addition, the distance between the light source and the lesion is fixed. This fixed distance must have a light intensity high enough for the beam energy to reach the lesion because of the distance difference [27]. Due to the high beam energy consumption, the module of the light source can lead to thermal destruction [27]. Therefore, a handheld type or light–weight fluorescence surgical microscope has an extremely low light energy intensity because the LED is also small [27]. 

This light energy intensity makes fluorescence emission guiding impossible. The only way to increase the light energy intensity is to use an LED with high light energy intensity or to use a laser [44]. However, as the size of the LED increases proportionally to the light energy intensity, there is a limit to the application of miniaturization and lightening. Furthermore, the unit price increases, and a large amount of heat is generated, which can lead to the thermal destruction of the module. The method using a laser has a light energy intensity that is more than twice as high as that of LED but is weak to heat and causes thermal destruction [27]. In addition, because lasers are harmful to the human body, the use of LEDs is recommended. A typical system may use a filter or beam splitter to separate the wavelength band of a light source and fluorescence emission. These optical components are not intended to increase energy but rather cause optical energy loss due to the loss of resistance to the material inside the optical components [27]. However, the proposed method increases the light energy intensity by a minimum of 1.5 times when using a mirror, regardless of the type of LED, and the unit price is 1/6, which is quite cheap. If packaging technology is used, miniaturization design is also possible.

## 5. Conclusions

Lasers are often used to observe the circulation of blood flow in cancer tissues or blood vessels via fluorescence emission during surgery. This is because they have a high beam irradiation power for sufficient fluorescence expression. However, the device can be damaged by heat due to their high use of energy. Additionally, the unit price is high owing to its complicated structure. Moreover, the narrow beamwidth of the laser hinders the investigation of the entire lesion. The beam width can be increased by connecting a separate beam expander to the laser. However, using this method, the laser power is reduced because of the resistance inside the splitter or expander, which causes difficulties in inducing fluorescence expression. In addition, lasers are harmful to the human body, and there are many restrictions on the process of licensing medical devices. LEDs can be used to avoid these problems; however, LEDs cannot satisfy the conditions of fluorescence expression because their power weakens as the beam width increases. The proposed quasi–symmetrical LED illumination method, which can increase power using a mirror, can generate four times more power than using only the LED itself. In addition, because the beam width is large, the entire area of the lesion is uniformly irradiated. When four lossless mirrors are used with one LED on top, a relatively large beam width can be ensured without refractive losses; therefore, the entire lesion can be irradiated, and the power can be amplified to sufficiently produce fluorescence. Furthermore, LEDs are expected to be harmless to the human body and advantageous for licensing medical devices. This study is very suitable for application in a surgical microscope for tumor removal and observation of the blood circulation state of blood vessels through fluorescent staining. Therefore, if the LED using the designed beam mirror is applied to a surgical microscope, it will be possible to sufficiently observe the blood circulation state of blood vessels and the tumor removal state by emitting fluorescence through high light source irradiation. Therefore, the proposed method is expected to be fully applicable in the operating room of surgeons in the future. If the proposed method is produced by adding a semiconductor process in the future, it could be mass–produced with ultrasmall modules. Additionally, it can be supported by miniaturization technology when applied to a surgical microscope.

## Figures and Tables

**Figure 1 diagnostics-13-02763-f001:**
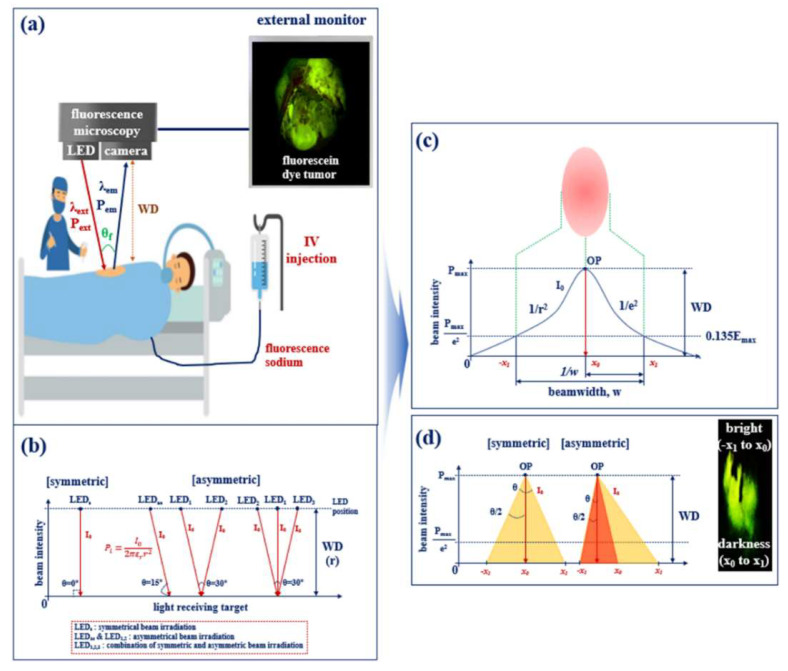
Definition of the single symmetric and asymmetric LED beam irradiation: (**a**) tissue–diagnosis–guided fluorescence emission in the surgery, (**b**) direction of the LED beam irradiation, (**c**) LED beam irradiation and brightness change rate according to slope, and (**d**) difference of the distribution and beam width through the symmetric and asymmetric LED.

**Figure 2 diagnostics-13-02763-f002:**
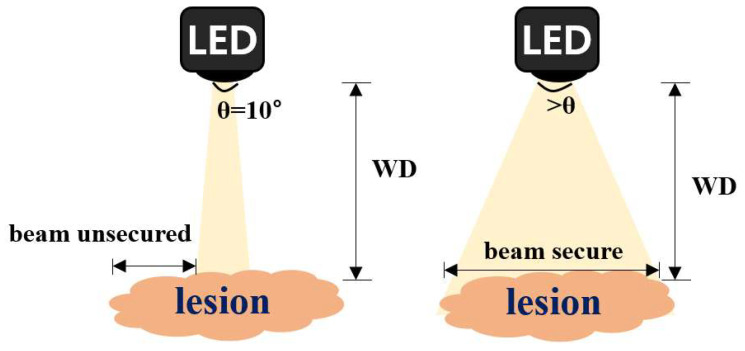
Difference of LED irradiation beams to secure the field of view of fluorescence–emission–guided lesion observation.

**Figure 3 diagnostics-13-02763-f003:**
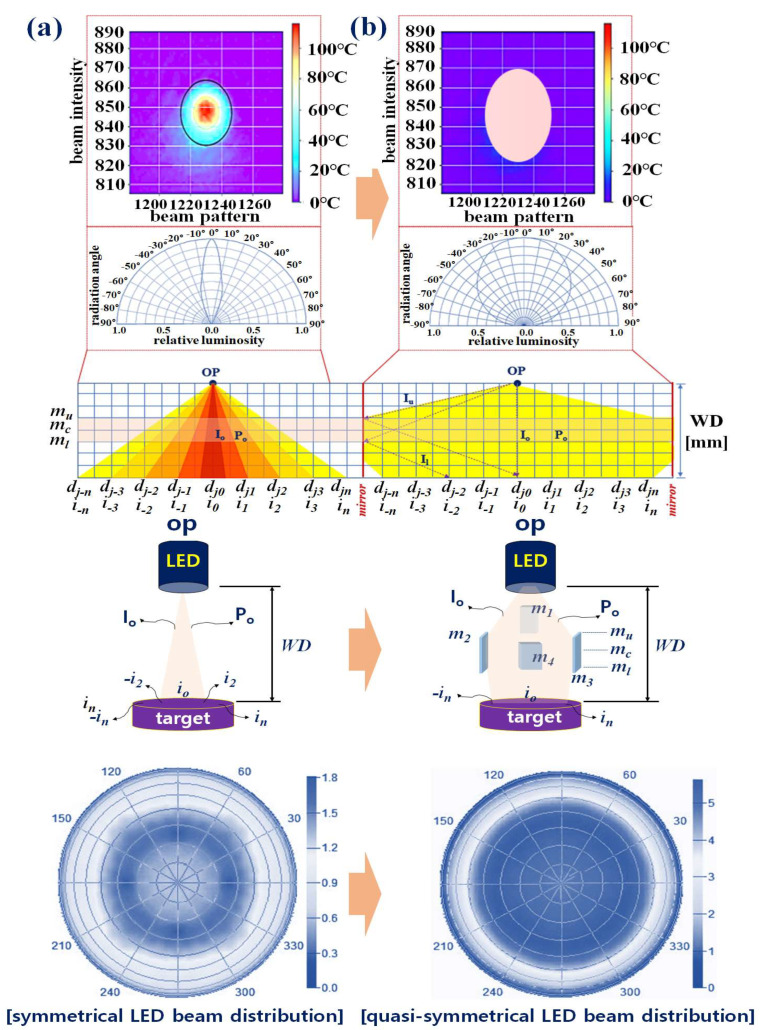
Characteristics of the symmetrical and quasi–symmetrical LED beam irradiation: (**a**) symmetrical LED concept and (**b**) quasi–symmetrical LED concept.

**Figure 4 diagnostics-13-02763-f004:**
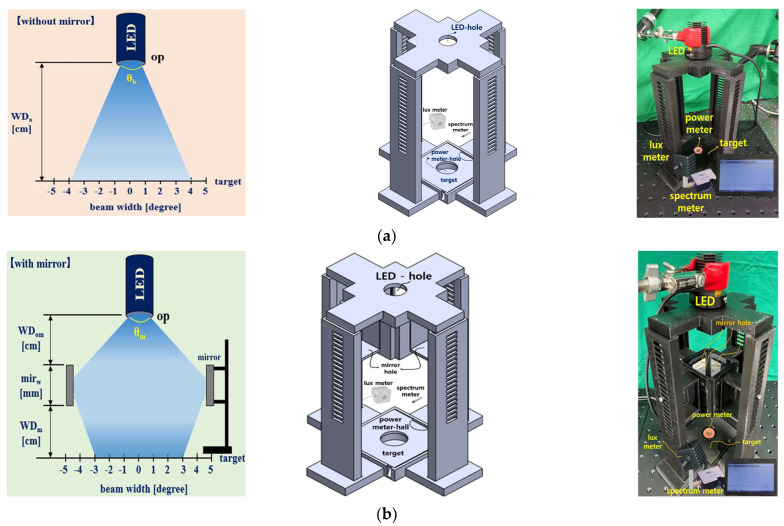
Concept for symmetrical and quasi–symmetrical LED beam irradiations: (**a**) conventional LED irradiation methods (without mirror), (**b**) quasi–symmetrical LED irradiation methods, (**c**) measurement of power with LEDs for conventional method and quasi–symmetrical LED irradiations, and (**d**) measurement with different numbers of mirror: m 1 to 4. (Refer to Appendix A for (**a**,**b**)).

**Figure 5 diagnostics-13-02763-f005:**
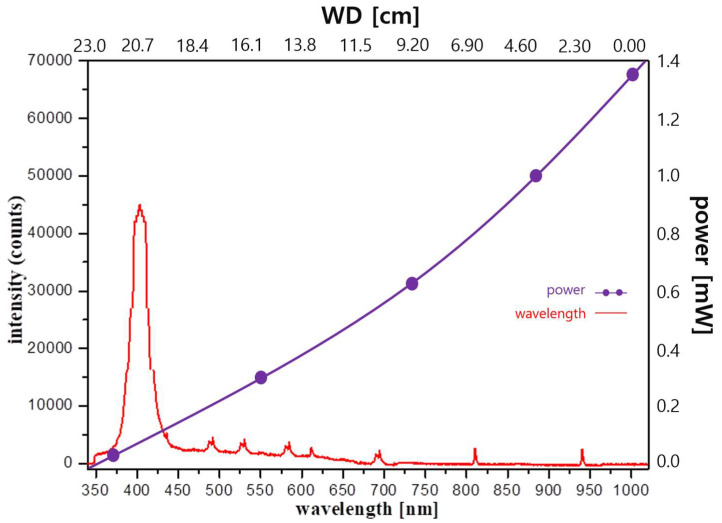
Measurement results of the irradiation beam using conventional method.

**Figure 6 diagnostics-13-02763-f006:**
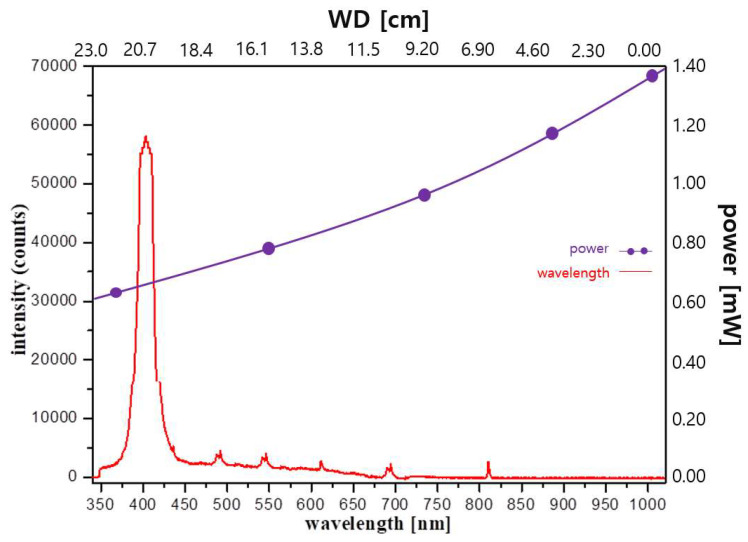
Measurement results of the quasi–symmetric LED beam irradiation method.

**Figure 7 diagnostics-13-02763-f007:**
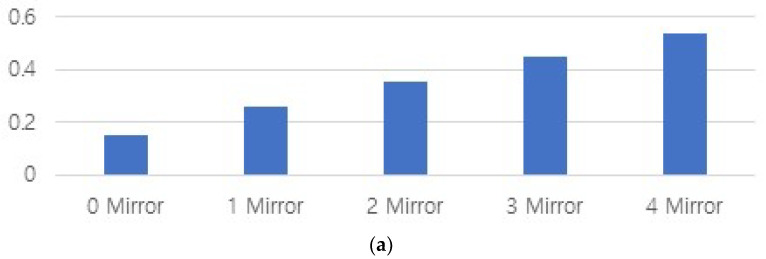
Test results of power (*P_o_*) and intensity (*I_o_*) change as the number of mirrors is varied: (**a**) differences graph and (**b**) power (mW) and intensity (lux) graphs.

**Figure 8 diagnostics-13-02763-f008:**
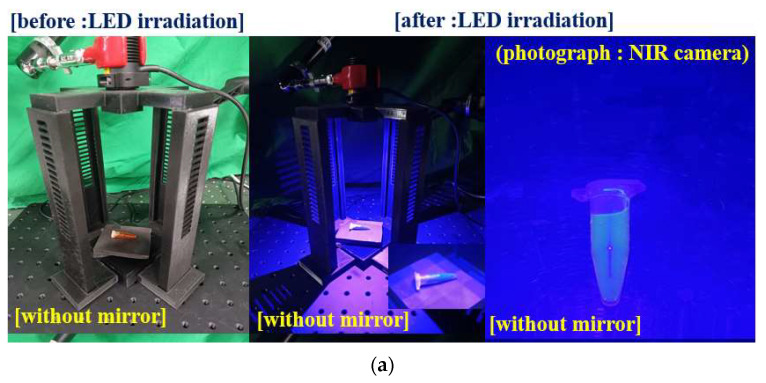
Fluorescence emission test for difference of performance with symmetrical LED and quasi–symmetrical LED using fluorescent sodium: (**a**) symmetrical LED test and (**b**) quasi–symmetrical LED test. (Refer to Appendix A).

**Figure 9 diagnostics-13-02763-f009:**
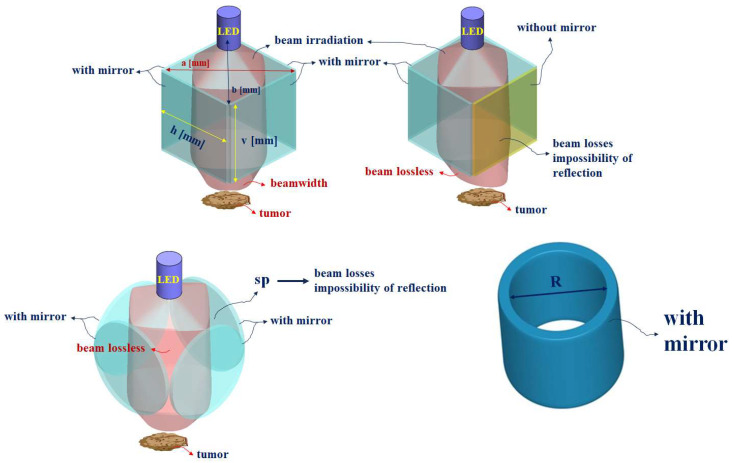
Characterization of LED beam irradiation using various mirror shapes.

**Table 1 diagnostics-13-02763-t001:** Measurement results according to mirror quantity.

Mirror Quantity	0	1	2	3	4
Light brightness change	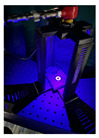	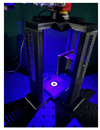	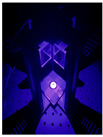	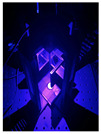	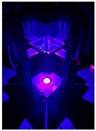
Power (*P_o_*) [mW]	0.15	0.258	0.352	0.448	0.600
Illuminance intensity [lux]	858	18,544	25,303	32,199	41,144

**Table 2 diagnostics-13-02763-t002:** Comparison for LED power with working distance (20 cm) of proposed and other methods.

Ref. [#]	λ_ext_ [nm]	WD [cm]	*P_max_* [mW]	*P_o_* [mW]	θ_w_ [degree]	Characteristic
[32]	467	6.17	100	6.10	30.0	LED
[40]	405	0.25	40	12.3	6.0	laser
[41]	405	0.0133	30	13.2	26	laser
[42]	405	35.0	4.70	6.00	42	laser
[43]	460	100	1540	12 µW	26.6	LED
this work	405	20.0	18.0	0.6	43	LED

**Table 3 diagnostics-13-02763-t003:** Comparison for LED performance of proposed method and others.

Ref. [#]	λ_ext_ [nm]	WD [cm]	*P_max_* [mW]	*P_o_* [mW]	θ_w_ [degree]	Characteristic
[32]	467	20	100	0.36	30.0	LED
[40]	405	20	40	1.60	6.0	laser
[41]	405	20	30	0.33	26	laser
[42]	405	20	4.70	0.38	42	laser
[43]	460	20	1.54 W	12 µW	26.6	LED
this work	405	20	18.0	0.6	43	LED

## Data Availability

The data presented in this study are available upon request from the corresponding author. The data are not publicly available because of privacy and ethical restrictions.

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
