# Peer review of "Single Quasi–Symmetrical LED with High Intensity and Wide Beam Width Using Diamond–Shaped Mirror Refraction Method for Surgical Fluorescence Microscope Applications"

_diagnostics, 2023, doi:10.3390/diagnostics13172763_

Round 1

Reviewer 1 Report

The authors present a novel method for application with surgical microscopy. This type of methodology is critical to medical science as fluorescence applications are advancing significantly within the field. Overall, the manuscript is presented well and the figures do a good job describing the findings. However, the English and grammar of the manuscript make is nearly impossible to follow what the authors are presenting.

Poor. Absolutely needs to be edited before accepting or publishing.

Author Response

Comments and Suggestions for Authors

Comment 0 :

The authors present a novel method for application with surgical microscopy. This type of methodology is critical to medical science as fluorescence applications are advancing significantly within the field. Overall, the manuscript is presented well and the figures do a good job describing the findings. However, the English and grammar of the manuscript make is nearly impossible to follow what the authors are presenting.

Comments on the Quality of English Language

Comment 0 :

Poor. Absolutely needs to be edited before accepting or publishing.

comprehensive answer:

Thank you for your positive review. English sentences were corrected with the help of a grammar correction expert. Thank you.

Reviewer 2 Report

In manuscript titled “Single quasi-symmetrical LED with high intensity and wide 2 beam width using diamond shaped mirror refraction method for surgical fluorescence microscope applications”, the authors propose to increase the beam width and power of LED by utilizing the quasi-symmetrical beam irradiation method. Generally, this innovative method and result is expected to be sufficient for fluorescence emission. However, some revision has to be conducted before it could be accepted for publication in IJMS.

Some comments:

1. In the introduction section, the author need to provide detailed information on current progress in the related field. The authors need to provide more information on the advantages of the relevant application in surgical fluorescence microscope.

2. Scientific and grammatical errors should be avoided. The current manuscript needs to be polished by a native English speaker.

3. Some more references related to fluorescent imaging and detection should be cited. For example,

S Xia et al ChemBioChem 20 (15), 1986-1994,

Y Zhang et al Microchemical Journal 180 (2022) 107619

Scientific and grammatical errors should be avoided. The current manuscript needs to be polished by a native English speaker.

Author Response

Comments and Suggestions for Authors

In manuscript titled “Single quasi-symmetrical LED with high intensity and wide 2 beam width using diamond shaped mirror refraction method for surgical fluorescence microscope applications”, the authors propose to increase the beam width and power of LED by utilizing the quasi-symmetrical beam irradiation method. Generally, this innovative method and result is expected to be sufficient for fluorescence emission. However, some revision has to be conducted before it could be accepted for publication in IJMS.

Answer:

Thank you for your positive feedback and comments. I made an effort to answer your opinion sincerely your opinion.

Some comments:

Comment 1 :

In the introduction section, the author need to provide detailed information on current progress in the related field. The authors need to provide more information on the advantages of the relevant application in surgical fluorescence microscope..

Answer 1 :

Thanks for the detailed comments. Lines 45-47 (yellow color) and 53-71 (green) have been added to the comments you requested.

Comment 2 :

Scientific and grammatical errors should be avoided. The current manuscript needs to be polished by a native English speaker.

Answer 2 :

Thank you for your positive review. English sentences were corrected with the help of a grammar correction expert. Thank you.

Comment 3 :

Some more references related to fluorescent imaging and detection should be cited. For example,

S Xia et al ChemBioChem 20 (15), 1986-1994,

Y Zhang et al Microchemical Journal 180 (2022) 107619.

Answer 3 :

I have added references to your request as follows.

S Xia et al ChemBioChem 20 (15), 1986-1994, ----à[7],

Y Zhang et al Microchemical Journal 180 (2022) 107619. ---à[8]

Also, I added to reference of [14]-[16], [22]-[32].

Please refer to manuscript.

Comments on the Quality of English Language

Scientific and grammatical errors should be avoided. The current manuscript needs to be polished by a native English speaker.

Answer :

Thank you for your positive review. English sentences were corrected with the help of a grammar correction expert. Thank you.

Reviewer 3 Report

1. In the scheme of increasing light intensity and improving uniformity, setting a lens group at the light source is a common scheme, and adding a comparison will be more convincing. And such a mirror group is more space, whether it will bring inconvenience to use.

 2. The author used four mirrors to achieve a good imaging effect, whether more mirrors will have a better effect, whether the use of cylindrical mirrors will have the best effect.

 3. In Figure 7b, the number of mirrors corresponds to a continuous test curve, where the points do not correspond to the number of mirrors, which sometimes makes understanding difficult.

Minor editing of English language required

Author Response

Comments and Suggestions for Authors

Comment 1 :

In the scheme of increasing light intensity and improving uniformity, setting a lens group at the light source is a common scheme, and adding a comparison will be more convincing. And such a mirror group is more space, whether it will bring inconvenience to use.

Answer 1 :

Thanks for your advice, Lenses are used to separate or pass a range of wavelengths. In order to increase the intensity of the beam, the source of the light source must be increased, but this method has several limitations. Therefore, this study used a mirror. Details are documented in the discussion session, lines 367-391 (green).

Comment 2 :

The author used four mirrors to achieve a good imaging effect, whether more mirrors will have a better effect, whether the use of cylindrical mirrors will have the best effect.

Answer 2 :

You have provided me with valuable ideas. thank you so much. At first, I worried a lot, but it turned out that 4 mirrors in a square shape are optimal. Here's why. And about sentences, the 335-366 (yellow) and presented in Figure 9. Thank you.

Comment 3 :

In Figure 7b, the number of mirrors corresponds to a continuous test curve, where the points do not correspond to the number of mirrors, which sometimes makes understanding difficult.

Answer 3 :

The dots on the graph are the division marks of the identification graph to classify the power lux. Symbols have been added so that the graph separation can be displayed well even in the black and white version. It has nothing to do with the number of mirrors. However, it can be confusing, so I changed the graph type to solid and dotted lines. Please check Figure 7.

Comments on the Quality of English Language : Minor editing of English language required

Answer: Thank you for your positive review. English sentences were corrected with the help of a grammar correction expert. Thank you.